# A Low-Cost Secure IoT Mechanism for Monitoring and Controlling Polygeneration Microgrids

**Josué Martínez-Martínez** [1] **, Diego Aponte-Roa** [1,*] **, Idalides Vergara-Laurens** [1] **and Wayne W. Weaver** [2]

1   Electrical and Computer Engineering Department, Ana G. Méndez University, Gurabo 00778, Puerto Rico; jmartinez718@email.uagm.edu (J.M.-M.); ivergara@uagm.edu (I.V.-L.)
2   Mechanical Engineering Department, Michigan Technological University, Houghton, MI 49931, USA; wwweaver@mtu.edu
*   Correspondence: aponted1@uagm.edu; Tel.: +1-787-806-8053

**Abstract:** The use of Internet-connected devices at homes has increased to monitor energy consumption. Furthermore, renewable energy sources have also increased, reducing electricity bills. However, the high cost of the equipment limits the use of these technologies. This paper presents a low-cost secured-distributed Internet of Things (IoT) system to monitor and control devices connected in a polygeneration microgrid, as a combined power system for local loads with renewable sources. The proposed mechanism includes a Wireless Sensor Actuator Networked Control System that links network nodes using the IEEE 802.15.4 standard. The Internet communication enables the monitor and control of devices using a mobile application to increase the efficiency. In addition, security mechanisms are implemented at several levels including the authentication, encryption, and decryption of the transmitted data. Furthermore, a firewall and a network intrusion detection-and-prevention program are implemented to increase the system protection against cyber-attack. The feasibility of the proposed solution was demonstrated using a DC microgrid test bench consisting of a diverse range of renewable energy sources and loads.

**Keywords:** IoT; cyber-physical system; renewable energy; microgrid; polygeneration

## 1. Introduction

According to the United States Energy Information Administration (EIA), over \$370 billion has been spent annually in recent years on electricity generation. Roughly 48% (\$177.6 billion) of the total generation was spent in the residential sector [1]. To increase the efficiency in energy consumption, data acquisition equipment such as smart meters, smart plugs, and many other sensors are commonly used. Although Home Energy Management System (HEMS) are anticipated as one of the promising technologies to fulfill the demand for energy saving, thus far it has only had limited implementation because the high initial equipment cost [2]  and the lack of optimal managing software because the power consumption pattern of residence is too diverse to find an optimal power management for each home [3]. A HEMS works by placing sensors at different appliances and/or devices that reads their energy consumption. By scheduling major household appliances, residents can reduce their electricity bills [4–6]. In several cases, the sensors are connected to the Internet reporting the data to the owner. The extension of the Internet connectivity into physical devices is named the Internet of Things (IoT), defined by the Institute of Electrical and Electronics Engineers (IEEE) as a network of items embedded with sensors that are connected to the Internet [7].

Monitoring applications can provide valuable information about your home from the actual status to a detailed history. This technology can provide the services, information, communication, and data

analysis anytime, as long there is a connection to the Internet, to tackle the energy consumption along with augmentation of the modern home living experience [8–10]. In addition, the integration of renewable energy sources (RESs) in distributed generation (DG), with proper control systems, have proven a lower electric bills in the residential sector [11]. In addition, the usage of RESs reduces carbon dioxide emissions by 1925 million metric tons in 2015, or roughly 36.5% of the total of all sectors in the United States [12]. However, the integration of RESs presents challenges in power quality, cost, power availability, location of RESs resource, and power forecast, among others [12]. Therefore, the integration of an IoT monitoring systems would improves the network performance to make a more resilient and efficient cyber-physical system. Since this environment is connected to the Internet, it needs to be protected from security threats. As shown in [13], the scheme has to fulfill three requirements:

1. Confidentiality concerns the protection of data, such that only approved users can access the data.
2. Authentication means checking that the data have not been altered and that the data can be confirmed by the claimed author to have been sent.
3. Access refers to only allowing suitably authorized users to access data, communications network, and computing resources and ensuring that those authorized users are not prevented from such access.

This paper presents a cyber-physical system consisting of a low-cost secure IoT mechanism for monitoring and controlling appliances/devices connected in a polygeneration microgrid. The hardware of the system was implemented using low-cost commercial off-the-shelf items, while the software was developed using free and open source resources. The IoT control scheme is comprised of a Wireless Sensor and Actuator Networked Control System (WSANCS) which implements a star topology with one sink node (S-N) and a set of sensor-actuator nodes (SA-N). The sensor-actuator nodes are an embedded entity composed of a micro-controller, a communication device, and a current/voltage sensor. The sink node is an embedded entity comprised of a micro-controller, a communication device, and a micro-computer. The data gathered by the WSANCS are sent to a cloud database. In addition, a mobile application allows the user to monitor and control the system in real-time. This mobile application retrieves the data from the database through the encrypted Transport Layer Security (TLS) protocol, providing to the system both the energy production and consumption. Finally, the mobile application allows the user to control all the registered electric appliances as well, and, according to the literature, these smart homes applications might achieve around 8% reduction in energy consumption [14,15]. Therefore, the use of these approaches in RES grids might represent a considerable improve in energy efficiency since the energy availability and storage are the bigger constraints in polygeneration microgrids.

This manuscript is organized as follows. Section 2 presents the related work. Section 3 presents the system architecture including the WSANCS architecture and details of the IoT secure architecture. Simulation results and security analysis are discussed in Section 4. Finally, Section 5 presents a summary and future work.

## 2. Related Work

A microgrid is a group of interconnected loads and distributed energy resources within a defined electrical boundaries [16]. Traditionally, microgrids have been used in remote areas without access to main power grids and can employ various communication technologies to data sharing. In addition, the governments in the U.S., the Asia Pacific region, and the European Union have established supporting policies, demonstration projects, control systems research, and the development of software tools for microgrids. For instance, The U.S. federal government provides investment tax incentives for customers installing microgrid technologies. The incentives cover a wide range of technologies including solar photovoltaics, combined heat and power, and electric vehicles. Some states provide incentive for microgrid projects to supply individual customers

or critical loads such as hospitals, first responders, and water treatment facilities, after natural disasters such as Hurricane Sandy [17]. However, given the intermittency and variability of RES, microgrid customers need mechanisms for controlling power demand depending on power availability. Therefore, the U.S. Department of Energy has identified several core areas for microgrid controls: (1) frequency control; (2) volt/volt-ampere-reactive control; (3) grid-connected-to-islanding transition; (4) islanding-to-grid-connected transition; (5) energy management; (6) protection; (7) ancillary service; (8) black start; and (9) user interface and data management [18].

Wireless Sensor Networks (WSN) are used in area monitoring, threat detection, the domestic sector, health care sector, environmental sector, industrial sector, and the primary sector. A WSN is defined as a self-configured and infrastructure-less wireless network of spatially dispersed sensors dedicated to monitoring and recording physical conditions of the environment [19]. A WSN can have a few to even thousands of nodes, where each node is connected to one or many sensor nodes to cooperatively pass their data across the network to a main location (usually a data center). Each sensor node is typically composed of a radio transceiver with an internal or external antenna, a micro-controller, an electronic circuit interfacing with the sensors and an energy source. WSNs topologies can vary from a simple star topology to a multi-hop wireless mesh topology [20]. Size and cost constraints on sensor nodes result in corresponding constraints on resources such as energy, memory, computational speed, and communications bandwidth. Moreover, a WSANCS is composed of a group of distributed sensors and actuators that communicate through wireless link which achieves distributed sensing and executing tasks [21]. This latter scheme allows the distributed control of devices that are spread in a wide area without physical connections between them such as a polygeneration distributed system with several renewable energy sources.

A WSANCS connected to the Internet becomes an IoT system. Coelho [22] illustrated the integration of the IoT for monitoring tasks in the office sector. They showed the convenience of having a monitoring system for conserving energy. A similar application was implemented by Patchava [23], who demonstrated the potentiality of having a smart home monitoring system. However, the use of this communication technology puts at risks the security and privacy of the data. IoT uses the Internet, thus carrying its vulnerabilities. These systems are vulnerable to diverse types of attacks such as denial of services, packet sniffing, and session hijacking to the confidentiality, integrity, and availability of the system, both damaging the system and putting the user to a possible personal risk [24,25]. In [26], Rawlinson described that 70% of the most popular IoT devices have vulnerabilities. Additionally, more than 25 vulnerabilities per device were discovered on average, with a total of 250 security concerns across all products tested. Furthermore, some devices had unsafe user interfaces and/or insecure firmware. Consequently, IoT devices need an efficient and effective protection scheme.

## 3. The Proposed System Architecture

This section presents the IoT WSANCS architecture implemented in a DC microgrid. The IoT security protection scheme is also introduced.

### 3.1. The IoT WSANCS Architecture

Figure 1 presents the updated system architecture introduced in [27]. The system consists of two RESs including a micro-wind turbine and a photovoltaic system (PV Array), both with their respective charge controller. It is worth noting that the architecture allows the system to increase the number of RESs as needed. The produced energy is stored in a battery bank. In addition, an inverter and a DC-to-DC converter are required to supply the loads. The WSANCS is composed by two types of nodes: the sensor-actuator nodes (SA-N) and the sink node (S-N).

The SA-N are included in every load of the system while the S-N is responsible for receiving the sensed data, process them, and control the network by sending wireless-control signals to the SA-N. The current and voltage are measured at different points in the microgrid using the SA-N. XBee modules are included in the WSANCS for wireless transmission purposes. Switches are connected to

the Arduino micro-controller to control the loads (turn-on/turn-off) through a digital signal. The SA-N architecture (seen in Figure 2) is composed of a current sensor, an Arduino micro-controller, and a XBee Pro module as end device using the IEEE 802.15.4 protocol [28] to monitor the grid generation and consumption. A voltage sensor is also included to measure the quality of voltage levels at different points in the microgrid.

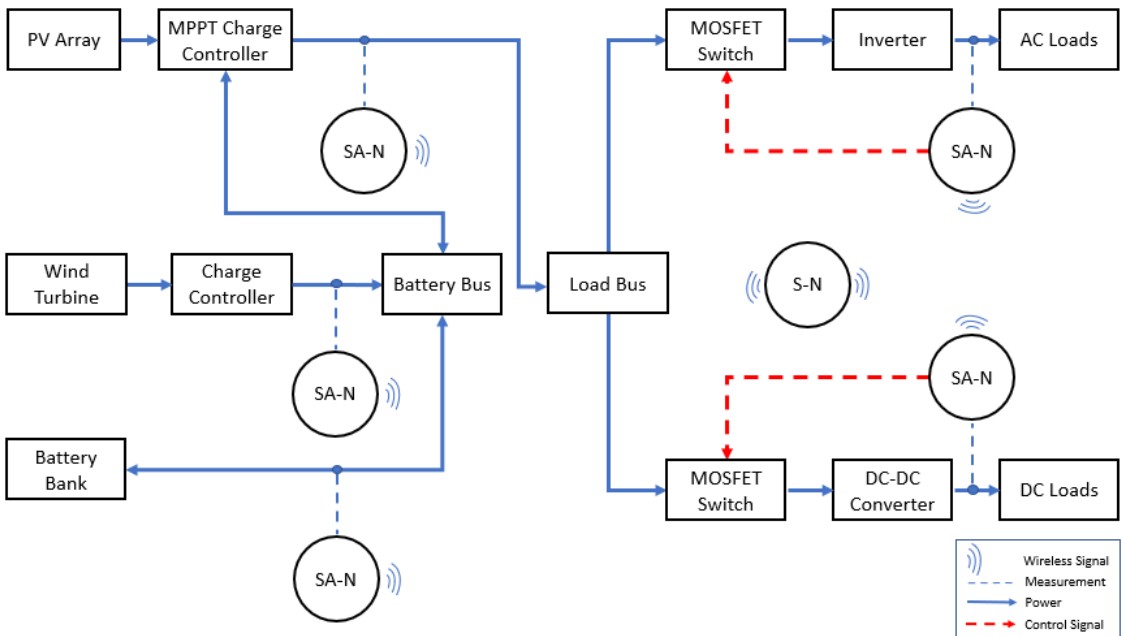

**Figure 1.** System architecture.

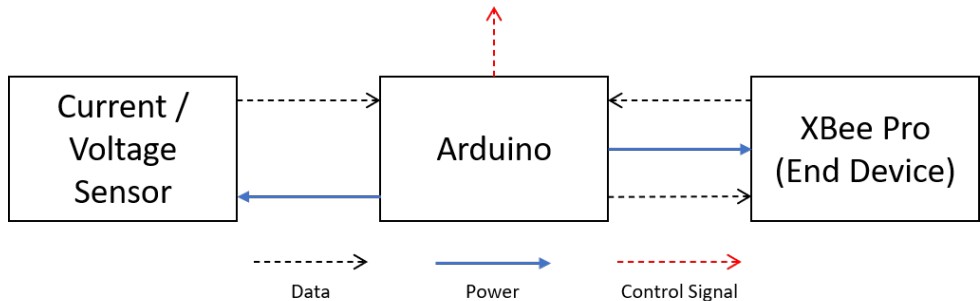

**Figure 2.** Sensor-actuator node architecture.

The S-N architecture (see Figure 3) includes the XBee coordinator connected to the Arduino 2 MEGA microprocessor for reading the signal of all the sensor nodes, making the necessary calculations, and printing those values to the corresponding serial port. A Raspberry-PI3 B+ (RPI3+) computer collects these values using a Python script to store data in both local and cloud databases using Sqlite3 and Firestore, respectively. This information is accessible for the user through the Android mobile application explained in detail in Section 3.2.3. In addition, the user can send the command to switch the state of the loads connected in the microgrid test bench through the mobile application, and then the S-N will send the signal to the corresponding SA-N to achieve what the user desires.

Since there are different XBees frequently attempting communication, the implemented module includes an acknowledgment scheme that allows the user to verify if data were received and from which device. The system was designed as a centralized coordinator-end device system with a coordinator XBee in the S-N that receives data from the SA-N and delivers them to a RPI3+.

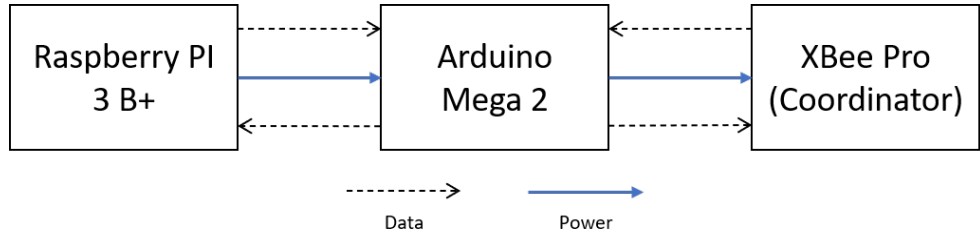

Data          Power

**Figure 3.** Sink node architecture.

### 3.1.1. Range and Operation Modes

The maximum distance to transmit data between two XBees was around 60 feet (18.3 m), taking into consideration concrete walls. The XBee's radio modules require 3.3 V with a power-down current < 10 μA, a transmit (TX) current of 45 mA, and a receive (RX) current of 50 mA. A consumption of less than 50 μA is expected in sleep mode operation; in this mode, for 1 min, the micro-controller keeps receiving information from the current or voltage sensors. Then, when the XBee wakes up, the acquired power consumption is sent to the sink node.

### 3.1.2. Network Topology

The WSANCS is implemented using a start topology because of its improved ability to contain errors and their tolerance to proximity and common-mode failures [29–33]. A quantitative comparison of the error-containment capabilities was presented by Barranco [34]. In the proposed topology, each SA-N has a XBee end-device used to connect it to the S-N which has the XBee coordinator. The XBee technology defines a point-to-point (PTP) connection between a coordinator and an end device. The coordinator receives a signal from any SA-N and can route them to the other SA-N. The S-N works as a server and it controls and manages entire function of the network. Some of the advantage of using this topology are:

1.  It is easy to locate problems because if an end device fails, it affects only one sensor node.
2.  It is easy to extend the network without disturbing the entire WSANCS.
3.  It is easy to identify faultS and remove nodes in the WSANCS.
4.  It provides very high speed of data transfer.

In addition, the S-N is connected to Internet through the RPI3+. This connection allows the system to send and received data to the user's mobile application as well as to upload the sensed data to a cloud database.

### 3.2. The IoT Security Protection Scheme

Khan in [35] described that the structure of IoT is divided into three main layers: (1) perception; (2) network; and (3) application. The large quantities of data generated by IoT devices, and their sensitive nature, make the IoT a prime target for cyber-attacks [36]. For this reason, the transmitted data are encrypted and authenticated using different protocols to secure the IoT layers [37].

Figure 4 presents the proposed protection for each IoT Layer. In the figure, red is assigned for the WSANCS nodes, blue for protocols, violet for intrusion detection systems, clear blue for databases, green for mobile applications, and orange for authentication measures. The implemented tools for every layer are explained below.

### 3.2.1. The Perception Layer

The perception layer is similar to the facial skin and five sense organs of the IoT, i.e., it mainly identifies objects and gathering information [38]. In this work, the perception layer is composed of WSANCS mentioned in Section 3.1. To assure data are traveling between the SA-N and the S-N, the IEEE 802.15.4 protocol authenticates the received message with the Media Access Control (MAC) address of the end devices.

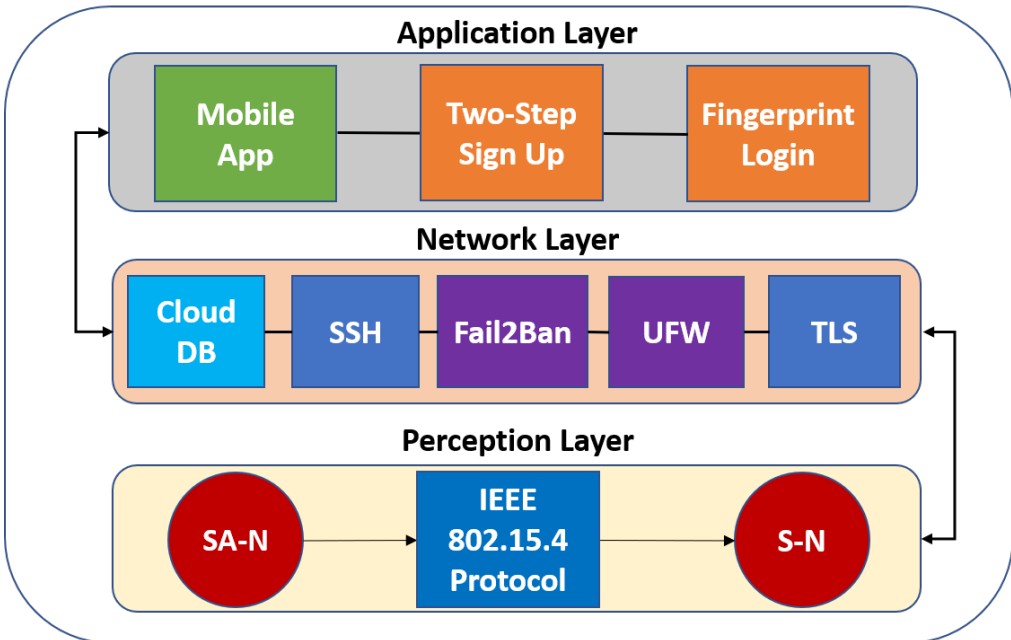

**Figure 4.** Proposed IoT security protection scheme.

### 3.2.2. The Network Layer

The network layer is for transmitting and processing data [38]. The network layer in this system includes the S-N. This node receives, processes, decrypts/encrypts data from/to the SA-N, and decrypts/encrypts data from/to the cloud-based database Firestore using the TLS protocol.

The major security threat is related to the access to the Internet. The Uncomplicated Firewall (UFW) is used to protect the RPI3+ against cyber-attacks, monitor network traffic, and block unauthorized traffic according to a defined set of security rules. Fail2Ban is used for intrusion prevention; this framework protects the S-N against brute force attacks and tracks the log files generated in order to detect malicious activity. Similarly, it can inform and update the firewall to avoid further suspicious IP login attempts. OpenSSH is used for providing a secured remote login connection to the RPI3+. This encrypts all traffic (including passwords) in order to prevent spying, hijacking, and other attacks, providing a secure network channel through a client–server architecture. The remote login is implemented using the key-based authentication standard where every authorized user must generate a key and encrypt the key with a passphrase to log on the computer. Finally, the passphrase is implemented in order to protect the user key in case a hacker takes control of the user computer and tries to log into the RPI3+ using the generated key.

### 3.2.3. The Application Layer

The application layer provides global management of the application based on the object's information processed in the database [35]. This layer is composed of a monitoring mobile application for Android. The security of this layer comes from all the security implemented in the perception and network layers and the authentication that is implemented in the mobile application to certify that is the user who is accessing the data and not an intruder. The application has a two-step authentication procedure to register a new user and a fingerprint authentication to log-in this user; both dashboards are shown in Section 4 of this paper.

Therefore, a home energy management mobile application for Android is developed. This application retrieves the consumption and generation data from the cloud database which allows the user to track their energy consumption and generation, providing information for scheduling some activities might demand more energy (e.g., use the washing machine when the PV and turbine are

generating more energy). A listener is implemented to refresh the consumption/generation graphs in real-time. In addition, the application lists all the monitored equipment with the option to be switched on and off to avoid energy waste when are being left on and nobody is at home. The user has the option to delete or register a new device by providing basic information about it as well. In addition, the application lists all the monitored equipment with the option to be switched on and off to avoid energy waste when they are left on and nobody is at home.

For security purposes, a two-step sign up verification was implemented with the user cell-phone number. After the sign up, the user has the option to log-in using his fingerprint if desired. Both features are explained in detailed in Section 4.4.1. The application dashboard and some capabilities are shown in Figure 5.

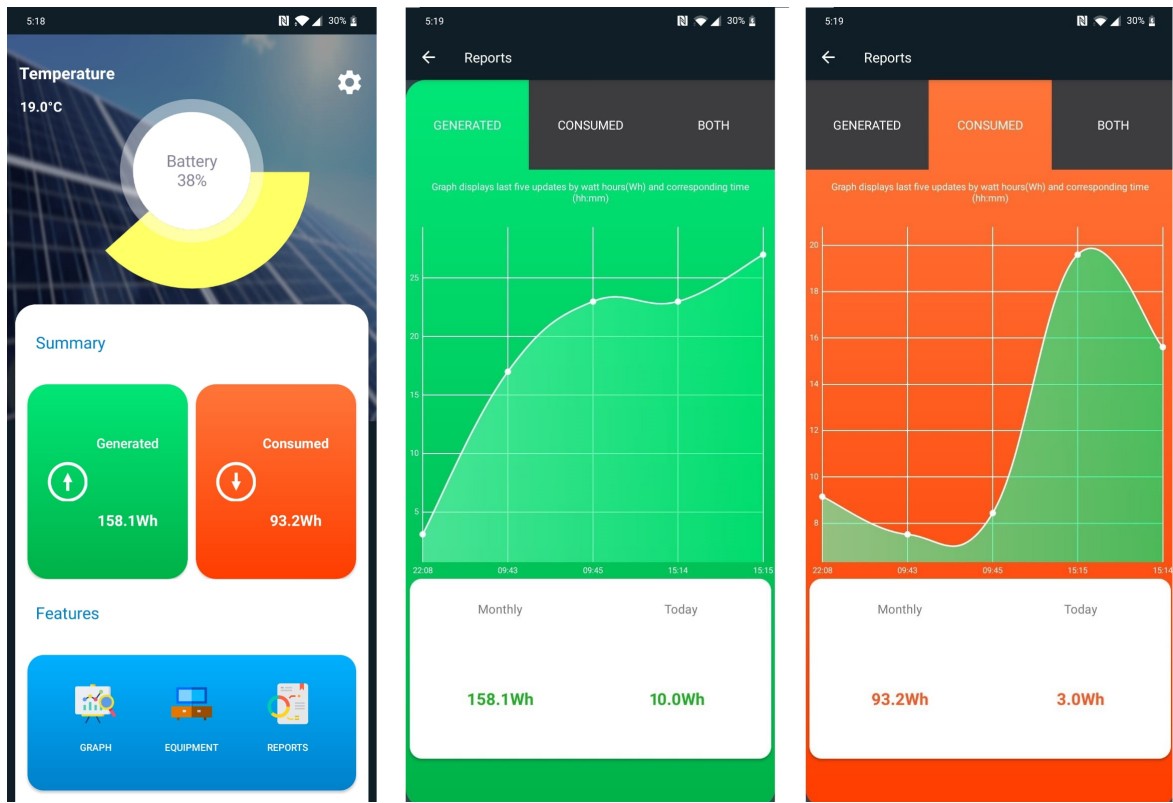

**Figure 5.** Mobile application capabilities.

Figure 6 shows the application interface listing the devices that the user has registered. The required information to register a new device is presented as well. A threshold value to be aware of an excess of power consumption could be included in order to turn-off the device when the power consumption reaches such threshold.

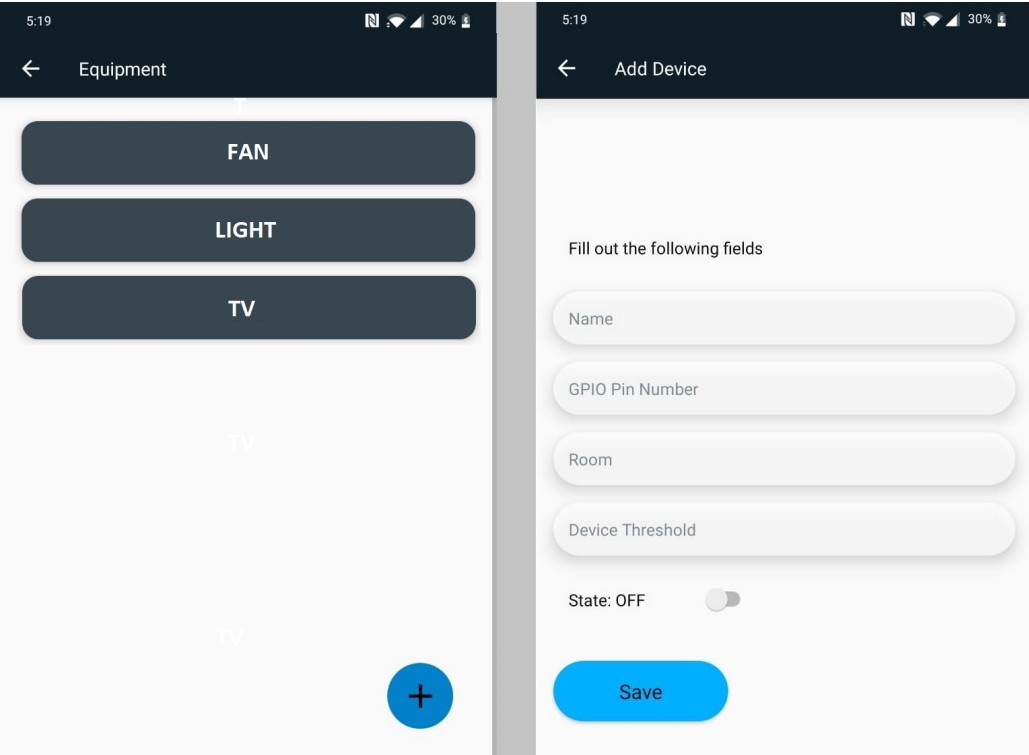

**Figure 6.** Some equipment options.

## 4. Results and Discussion

This section provides the simulation results and security analysis implemented in the DC microgrid test bench used as a test platform.

### 4.1. Microgrid Test-Bed

The proposed system was tested using the updated configuration of a DC microgrid test bench introduced in [27]. The test bed microgrid consists of two RESs including a 400 W micro-wind turbine and a 400 W PV system, each with its respective charge controller. The generated energy is stored in a 12 V deep-cycle battery bank. A 400 W inverter and a DC-to-DC converter were required to supply the loads. Arduino off-the-shelf modules are used to measure currents and voltages at different points in the microgrid, as presented in Figure 1. The WSANCS communication was implemented using the XBee modules. The mosfet switches are connected to the Arduino micro-controller to control the load state (turn-on or turn-off) through a digital signal, as shown in Section 3.

### 4.2. WSANCS Simulation

The Riverbed Modeler Academic Edition was selected to simulate the data transfer protocol used by the XBees. The simulation used the ZigBee protocol based on the IEEE 802.15.4 standard, and the data link layer worked with the same standard. The star topology presented in Section 3.2.2 was implemented.

To assure that the bidirectional data transfer is between the S-N and the SA-N, two SA-N were simulated. Figure 7 shows the model implemented were the node_0 is the coordinator and the other two nodes are end devices. The results demonstrate that the protocol can send 120 package in 1 min between these three nodes. A total of 1–60 package per minute is estimated for the desired application, which validates that protocol has the transfer capacity required.

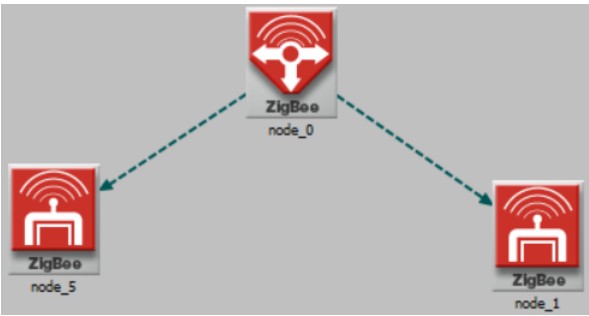

**Figure 7.** Communication simulation.

### 4.3. WSANCS Implementation

In a survey conducted by Lim [39], they found that 68.75% of 35 survey respondents occasionally forget to switch off unused appliances while 6.25% of the respondents tend to forget frequently. Besides that, it is also noted that 48.57% (17 respondents) of survey takers find it troublesome to walk to the switch to turn off an appliance, hence 35.29% of the above 17 respondents often leave the appliance turned on. To mitigate this waste of energy, a script was written with a listener to the Firestore database to perform real-time changes in the microgrid through the mobile application. The desired switch state can be uploaded by the user from the application to the cloud database. Then, the listener in the S-N uploads and downloads these changes. Section 3.1.1 mentions that the SA-N only wakes up every 1 min to send data. However, the S-N is able to send an interrupt signal to the SA-N to switch the state of the desired device instantly, after a user requested.

### 4.4. Security Protection Analysis

#### 4.4.1. Two-Step Verification and Fingerprint

Figure 8 presents the login interface of the mobile application. To achieve a higher level of security, the application contains a two-step verification process to create the user account. In this case, the two-step verification is implemented using the user phone number. Then, the user needs to enter the code that was sent by SMS. The authentication process is using SHA-256 to encrypt and authenticate such process. Log-in via fingerprint capability is available for better security if desired. A reset password interface is included as well. An email with detailed procedure will be received after a request.

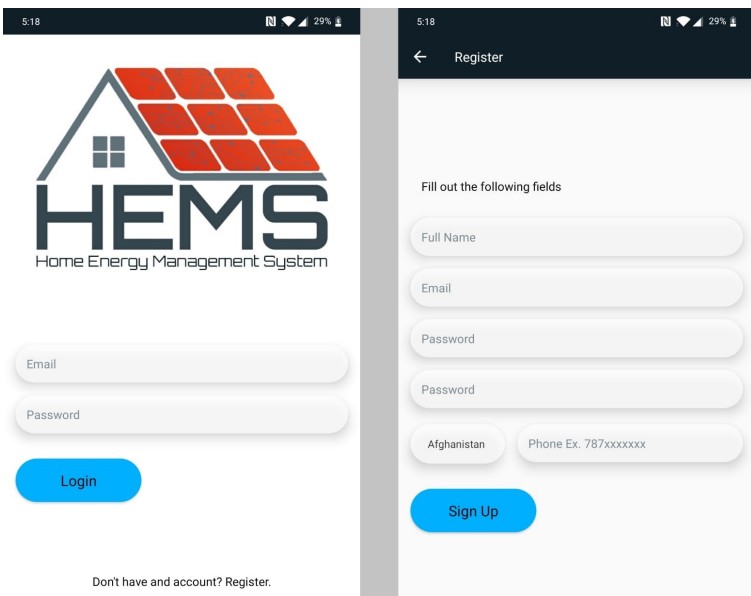

**Figure 8.** Login interface.

### 4.4.2. Brute Force Attacks

The penetration test was conducted with the Metasploit software that it is owned by Boston, Massachusetts-based security company Rapid7 [40]. The exploit was successful the first time in the insecure environment, which also has a Secure Shell (SSH) for secure remote login (SRL) but with a common password (e.g., abcdefg). This exploit provides the attacker with access to device management, as shown in Figure 9. There is a constant loop that the hacker can have access all the time. The SSH-key based protocol was used to secure the RPI3+ running Raspbian operating system. However, SSH-key-based protocol does not guarantee that the system will be resistant to such attacks but only those with the key and passphrase will be able to log-in and execute the exploits.

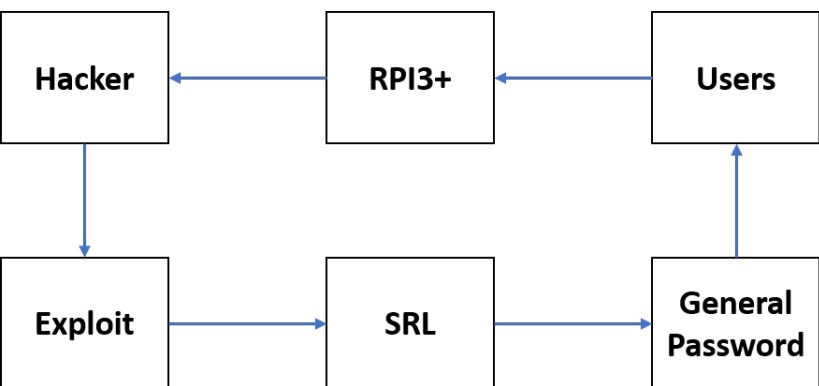

**Figure 9.** Pen-Test in vulnerable environment.

In addition, another attack was successfully performed because the user inserted the same general passphrase to encrypt the key. Taking into consideration this situation, the Fail2Ban and UFW installation settings were modified as well as the default rules. In addition, the key's passphrase was changed with numbers and special characters for the strongest one with more than 16 characters. Now, the exploit was run again, but the key's passphrase was hard to guess and after five tries Fail2Ban banned the IP address updating its Firewall's IP-Tables for an unlimited period. This procedure is shown in Figure 10. Then, the host files were modified for allowing only the IP address from the authorized users in the system. Thus, only registered user IP address can target and break the machine.

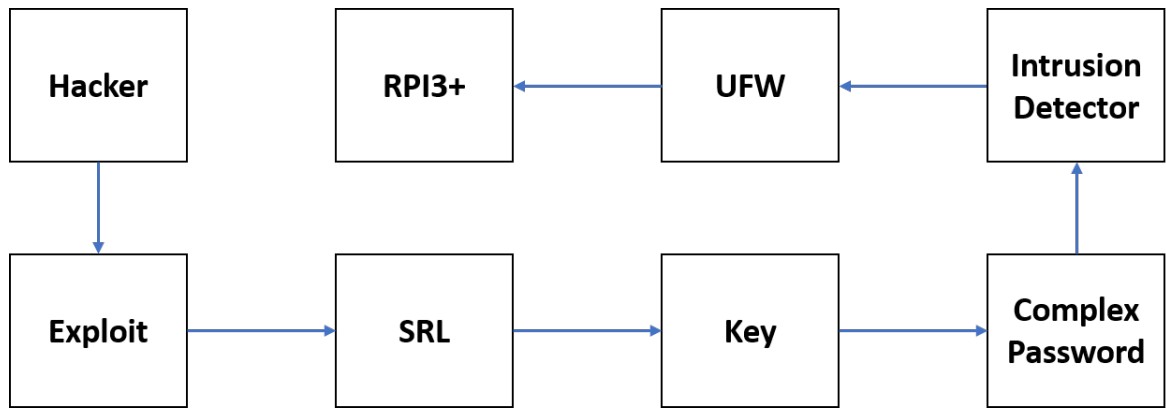

**Figure 10.** Pen-Test in the proposed scheme.

### 4.4.3. Denial of Service (DoS) Attacks

Low Orbit Ion Cannon (LOIC) [41] was the mechanism used to strike the RPI3+. One of the DoS attacks requests 530,020 to the server from which 1239 failed in vulnerable environment. A high number in the failed parameter means that the server goes down and does not respond. To mitigate this attack, modifications in the host (.allow and .deny) files and Fail2Ban (.local) rules were made.

After those changes, a total of 44,045 requests were made to the server by the first attack, and only 10 failed. In addition, the request to the server stops when the failed goes to 10 because the Fail2Ban detects and bans the invalid IP address. Then, another threat was carried out, since the Fail2Ban had blocked the IP address, 0 requests were processed and 0 failed, as shown in Figure 11. This indicates the IP address is blocked and can no longer attack or attempt to access the RPI3+.

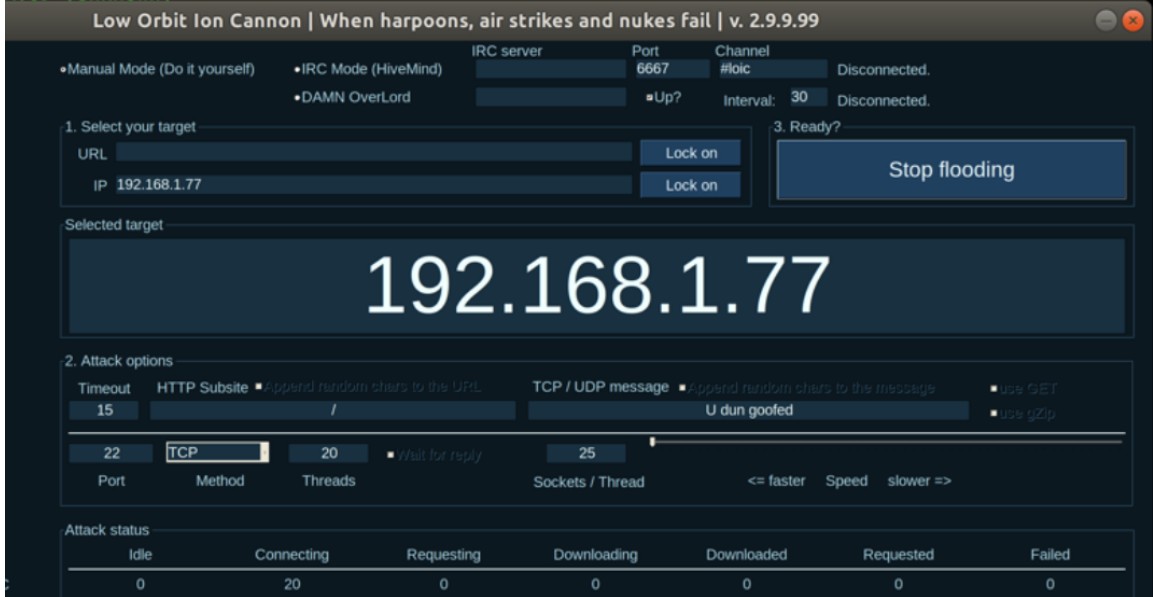

**Figure 11.** DoS stopped.

The calculated failure rate of our system after some runs of DoS attacks in the vulnerable environment was 0.20% but the CPU was working at 100%. In the case of DDoS attacks, the system can shut down for an amount of time without the proposed scheme in this paper. The failure rate equation used was:

$$F_{rate} = \frac{\sum_{i=1}^{n} F_i / R_i}{n} (100\%) \tag{1}$$

where $F$ is the failed parameter and $R$ is the requested parameter of the LOIC software.

## 5. Summary and Future Work

This paper presented a low-cost, secure IoT mechanism for monitoring and controlling appliances/devices in a microgrid with renewable energy sources. The integration of a WSANCS to the microgrid provides real-time access to each device connected to the system for monitoring and control purposes. In addition, a mobile application allows the user to visualize real time power consumption/generation data with a graphical user interface. As soon one of the categories is selected, a graph of the desired measurement is refreshed with the latest desired data, providing updated results. The application alerts the user when a device reaches a power consumption threshold defined by the user.

A control and security analysis was conducted to establish the feasibility of the proposed system. The results demonstrate the systems capabilities in a DC microgrid test bench. Brute force and denial of service attacks were mitigated involving free and open-source software mechanisms in the network layer of the IoT implementation.

Future work includes the increasing the microgrid size and developing an iOS version app to reach more users. In addition, the system can be improved through the use of machine learning algorithms to detect anomalies in the system or a turned-on device without human interaction based on previous data [42].

**Author Contributions:** Conceptualization, J.M.-M. and D.A.-R.; formal analysis, J.M.-M., D.A.-R., and I.V.-L.; funding acquisition, D.A.-R. and W.W.W.; investigation, J.M.-M. and D.A.-R.; methodology, J.M.-M., D.A.-R., and I.V.-L.; Project administration, J.M.-M. and D.A.-R.; resources, J.M.-M., D.A.-R., and I.V.-L.; software, J.M.-M. and I.V.-L.; supervision, J.M.-M., D.A.-R., and I.V.-L.; validation, J.M.-M., D.A.-R., and I.V.-L.; writing—original draft, J.M.-M. and D.A.-R.; and writing—review and editing, J.M.-M., D.A.-R., I.V.-L., and W.W.W. All authors have read and agreed to the published version of the manuscript.

**Funding:** This research was funded by Consortium for Integrating Energy System in Engineering and Science Education (CIESESE), a program supported by the U.S. Energy Department (DE-NA0003330).

**Acknowledgments:** The authors are grateful for the support of the José Domingo Pérez Engineering School at Ana G. Mendez University, Gurabo Campus, and to the Puerto Rico Energy Center (PREC). In addition, the authors gratefully acknowledge the contributions of the undergraduate students Javier Sanchez, Jorge Cruz, and Shervin Firouzdehghan to this paper.

**Conflicts of Interest:** The authors declare no conflict of interest.

## Abbreviations

The following abbreviations are used in this manuscript:

| | |
|---|---|
| CAN | Controller Area Network |
| DC | Direct Current |
| DG | Distributed Generation |
| MPPT | Maximum Power Point Tracking |
| HEMS | Home Energy Monitoring System |
| IoT | Internet of Things |
| MAC | Media Access Control |
| PTP | Point-to-Point |
| RESs | Renewable Energy Sources |
| RPI3+ | Raspberry Pi 3 B+ |
| SA-N | Sensor-Actuator Nodes |
| S-N | Sink Node |
| TLS | Transport Layer Security |
| UFW | Uncomplicated Firewall |
| WSANCS | Wireless sensor and Actuator Networked Control System |
| WSN | Wireless Sensor Network |

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
