# Peer review of "A Low-Cost Secure IoT Mechanism for Monitoring and Controlling Polygeneration Microgrids"

_applsci, doi:10.3390/app10238354_

Round 1

Reviewer 1 Report

"So far it has only had limited implementation
22 because the high initial equipment cost " also because of little impact unless the user is really interested and uses it as a tool, there are German studies showing that smart meters only work for technically minded interested people who can understand the information provided, please comment

please use also literature like https://www.isi.fraunhofer.de/content/dam/isi/dokumente/sustainability-innovation/2011/WP06-2011_smart-metering-in-Germany.pdf

"a home energy management mobile application for Android" what can this really do apart from being a nice toy, please comment

nice description of a nice system, to me it looks targetting interested consumers (which I consider the vast minority); how can a "normal" person use the system and what would be the benefits (switching equipment on or off, reducing the load on the grid), what could be the impact on these informed decisions, please elaborate.

As such the paper is the description of an Arduino/Raspberry solution, which apparently is not too challinging

Reviewer 2 Report

Interesting work. Minor editing corrections needed (i.e. line 105 - two "and"). I would also suggest to add a short (3-5lines) description of the SA-N especially as a device which measures current/voltage: what type of sensor, why it is needed to measure voltage (in the sense that operationg voltage is known, and therefore only the measurement of current is suffiecient for determining the power consumption).

Round 2

Reviewer 1 Report

The rebuttal table is not really answering the questions. Can the authors give at least a few examples of actual implementation of the system and lessons learned (reception by households, actual benefit in terms of energy savings or load shifting). Toys like this are around and the technical and scientific value of this exercise still need to be demonstrated. Examples: https://www.verbraucherzentrale.de/wissen/energie/erneuerbare-energien/energiemanagementsystem-fuer-zu-hause-energie-effizienter-nutzen-48095 (you probably need to use Google translate)

https://www.shine.eco/2017/01/30/was-bringt-energiemanagement-fuer-privathaushalte/

https://www.sems.energy/einsatz/

https://www.researchgate.net/publication/307668003_Smart_Energy_Management_for_Households

https://www.amazon.com/Smart-Energy-Management-Households-developers/dp/1492764779
